# Hydroxychloroquine Enhances Cytotoxic Properties of Extracellular Vesicles and Extracellular Vesicle–Mimetic Nanovesicles Loaded with Chemotherapeutics

**DOI:** 10.3390/pharmaceutics15020534

**Published:** 2023-02-05

**Authors:** Sergey Brezgin, Anastasiya Kostyusheva, Natalia Ponomareva, Ekaterina Bayurova, Alla Kondrashova, Anastasia Frolova, Olga Slatinskaya, Landysh Fatkhutdinova, Georgy Maksimov, Mikhail Zyuzin, Ilya Gordeychuk, Alexander Lukashev, Sergey Makarov, Alexander Ivanov, Andrey A. Zamyatnin, Vladimir Chulanov, Alessandro Parodi, Dmitry Kostyushev

**Affiliations:** 1Laboratory of Genetic Technologies, Martsinovsky Institute of Medical Parasitology, Tropical and Vector-Borne Diseases, First Moscow State Medical University (Sechenov University), 119991 Moscow, Russia; 2Division of Biotechnology, Sirius University of Science and Technology, 354340 Sochi, Russia; 3Chumakov Federal Scientific Center for Research and Development of Immunobiological Products, Russian Academy of Sciences (Polio Institute), 108819 Moscow, Russia; 4Institute of Molecular Medicine, First Moscow State Medical University (Sechenov University), 119991 Moscow, Russia; 5Faculty of Biology, Lomonosov Moscow State University, 119991 Moscow, Russia; 6School of Physics, ITMO University, Lomonosova 9, 191002 St. Petersburg, Russia; 7Qingdao Innovation and Development Center, Harbin Engineering University, Harbin 150001, China; 8Qingdao Innovation and Development Center, Harbin Engineering University, Qingdao 266000, China; 9Engelhardt Institute of Molecular Biology, Russian Academy of Sciences, 119991 Moscow, Russia; 10Faculty of Health and Medical Sciences, University of Surrey, Guildford GU2 7X, UK; 11Department of Infectious Diseases, First Moscow State Medical University (Sechenov University), 119991 Moscow, Russia

**Keywords:** exosomes, extrusion, proteolipid nanoparticles, drug delivery, nanomedicine, doxorubicin, extruded nanoparticles, hydroxychloroquine, endosomal escape, lysosomotropic agent

## Abstract

Because of their high biocompatibility, biological barrier negotiation, and functionalization properties, biological nanoparticles have been actively investigated for many medical applications. Biological nanoparticles, including natural extracellular vesicles (EVs) and synthetic extracellular vesicle–mimetic nanovesicles (EMNVs), represent novel drug delivery vehicles that can accommodate different payloads. In this study, we investigated the physical, biological, and delivery properties of EVs and EMNVs and analyzed their ability to deliver the chemotherapeutic drug doxorubicin. EMNVs and EVs exhibit similar properties, but EMNVs are more effectively internalized, while EVs show higher intracellular doxorubicin release activity. In addition, these nanotherapeutics were investigated in combination with the FDA-approved drug hydroxychloroquine (HCQ). We demonstrate that HCQ-induced lysosome destabilization and could significantly increase nanoparticle internalization, doxorubicin release, and cytotoxicity. Altogether, these data demonstrate that, from the delivery standpoint in vitro, the internalization of EMNVs and EVs and their payload release were slightly different and both nanotherapeutics had comparable cytotoxic performance. However, the synthesis of EMNVs was significantly faster and cost-effective. In addition, we highlight the benefits of combining biological nanoparticles with the lysosome-destabilizing agent HCQ that increased both the internalization and the cytotoxic properties of the particles.

## 1. Introduction

Bioinspired and biological nanoparticles (NPs) are currently under intense investigation because of their potential for drug delivery [1]. Using biological NPs in practice is particularly relevant due to their high biocompatibility, safety, stability, and ability to cross biological barriers, and the potential to program the cargo loading and/or targeted delivery of the system [2]. Within the large portfolio of biological carriers [3], extracellular vesicles (EVs), one of the most investigated platforms in the field [4], were recently tested in clinical trials for cancer treatment [5]. EVs are proteolipid NPs derived from the endosomal compartment and secreted by the cell after fusion with multivesicular bodies. These vesicles can provide the same biological identity as the cell source, including targeting properties [6]. EVs can be synthesized by almost any cellular phenotype [7,8,9] and can be loaded with different payloads, including small molecules [1] and biologics [10]. The major drawbacks of EVs reside in their expensive and lengthy production and in the complexity of postsynthetic manipulation to accommodate the payload while preserving the particle structure [11]. EVs have been investigated as drug delivery vehicles since the 1960s and are considered the gold standard of this kind of technology and immunologically safe [12].

On the other hand, extracellular vesicle–mimetic nanovesicles (EMNVs) are synthesized via cell extrusion through a series of filters with decreasing pore diameter [13]. This process results in carriers defined by a cell membrane enclosing the cytosol. Similar to EVs, EMNVs may derive from virtually any cell type [13]. However, unlike EV production, EMNV manufacturing does not require long-term cell culturing and large amounts of cell culture media. Their synthesis is independent from the cell state and their synthetic yield (particle/cell ratio) remarkably exceeds that of EVs [14,15]. Still, the practical use of EMNVs should be considered with caution because of potential contamination with unwanted cellular materials (e.g., nucleic acids) that can eventually affect their immunological tolerance [16]. Considering the inherent differences in EV and EMNV synthesis, it is reasonable to expect that their delivery properties could change both in terms of loading yield, internalization, and cytostatic properties.

In an effort to compare their delivery properties, it is worth consideration that, as with many other nanodelivery systems, biological nanoparticles can be sequestered in the endosomal compartment, affecting the overall efficacy [17]. One strategy to avoid endosomal sequestration is the use of hydroxychloroquine (HCQ), an antimalarial FDA-approved drug known for over 60 years in the clinic [18]. HCQ has lysosomotropic properties inhibiting lysosomal acidification, osmotic pressure, and P-gp activity [19]. Indeed, HCQ demonstrated increased cytotoxicity when combined with several synthetic nanotherapeutics [20,21].

In this work, we compared in vitro EMNVs’ delivery properties with EV formulations. Chemotherapeutic doxorubicin (DOX) was used as a payload model since it is extensively tested in the field [22]. For the first time, we also introduced an additional variable represented by combinatorial treatments with HCQ to test if this molecule could affect the internalization of nanoparticles, the release of DOX, and the cytotoxic activity of nanoformulations. NP internalization and DOX release (alone or in combination with HCQ) were first evaluated in widely used HEK-293T cells [23]. The cytotoxic effects of the selected drug formulations were then measured in different cell lines.

We discovered that although EMNVs and EVs have similar physical properties, EMNVs are slightly more efficiently internalized but are less effective in releasing DOX compared to EVs. DOX-loaded EMNVs and EVs showed similar cytotoxicity, and both the carriers were more effective than free DOX. HCQ favored endosomal destabilization and increased particle internalization and cytotoxic activity, highlighting the potential of this therapeutic in enhancing the delivery properties of biological NPs.

## 2. Materials and Methods

### 2.1. Cell Culture

HEK-293T, human hepatoma cells HepG2, and human breast cancer cells SKBR3 were cultured in DMEM (PanEco, Moscow, Russia) supplemented with 10% fetal bovine serum (FBS) (Cytiva, Logan, UT, USA), 2 mM of L-glutamine, 100 U/mL of penicillin, and 100 µg/mL of streptomycin (Gibco, Thermo Fisher Scientific, Oxford, UK). Cells were cultured at 37 °C and 5% CO_2_. For DOX internalization experiments and experiments with the caspase reporter system, HEK-293T cells were seeded onto 12-well plates at ~60% confluency. Caspase 3 activation was analyzed in HEK-293T cells transfected with ZipGFP-Casp3 plasmid (ZipGFP-Casp3 was a gift from Xiaokun Shu; Addgene #81241) [24]. Transfection was performed as follows: plasmid DNA was added to NaCl solution (solution A). Solution B containing polyethylenimine in NaCl was prepared in parallel, incubated for 10 min, and gently mixed with solution A. The combined solutions were incubated at room temperature for 10 min and then added to cells. The day after transfection, cell culture medium was discarded, and the cells were gently washed twice in PBS before fresh complete medium was added. For cell viability analyses, cells were seeded onto 96-well plates with 30% confluency.

### 2.2. Isolation of EVs

For isolating EVs, HEK-293T cells were conditioned in DMEM media supplemented with 10% EV-free FBS. FBS was depleted of EVs by ultrafiltration using Amicon Ultra-15 100 kDa filter devices (Merck Millipore, Darmstadt, Germany) as described previously [25]. The isolation of EVs from conditioned media was performed according to the protocol described by Heath et al. with modifications [26]. Conditioned media were consequently centrifuged at 300× *g* (10 min), 2000× *g* (10 min), and 10,000× *g* (10 min). Clarified media were applied onto columns filled with Macro-Prep DEAE Resin (Bio-Rad, Hercules, CA, USA), washed with 20 column volumes (CV) of buffer containing 50 mM of HEPES and 100 mM of NaCl, washed with 10 CV of buffer containing 50 mM of HEPES and 335 mM of NaCl, and eluted with 50 mM of HEPES/890 mM of NaCl buffer. Eluate was concentrated using Amicon Ultra-15 (100 kDa) filter devices (Merck Millipore) followed by 3 washes with PBS and reconcentration. Aliquots of isolated vesicles were lysed with RIPA buffer and diluted in PBS; total protein in samples was measured with a Pierce Coomassie (Bradford) protein assay kit (Thermo-Fisher Scientific, Waltham, MA, USA).

### 2.3. Preparation of EMNVs

HEK-293T cells (~5 × 10^6^) were washed twice with PBS and detached using Versene solution (PanEco). The cell suspension was serially extruded 7 times through 10-, 5-, and 1-µm polycarbonate membrane filters (Nuclepore, Whatman, Inc., Clifton, NJ, USA) using mini-extruders (Avanti Polar Lipids, Birmingham, UK). The resulting solution was centrifuged for 10 min at 2000× *g*, then for 10 min at 10,000× *g* to discard cell debris; the supernatant was then filtered via Centrisart 1 (300 kDa) Concentrator (Sartorius, Goettingen, Germany) followed by 3 washes with PBS and a reconcentration step. Aliquots of isolated vesicles were lysed with RIPA buffer and diluted with PBS, and total protein in samples was measured with a Pierce Coomassie (Bradford) protein assay kit (Thermo-Fisher).

### 2.4. Transmission Electron Microscopy

Transmission electron microscopy (TEM) assay of EV and EMNV samples was performed as described previously [26]. Before measuring, 5 µL of the EV/EMNV samples was pipetted onto a 400-mesh copper grid with carbon-coated formvar film and incubated for 1 min. Excess solution was removed by blotting with a filter paper, and the grid was rinsed in distilled water for 10 s. The grid was placed in 10 µL of 2% uranyl acetate, followed by blotting to remove excess liquid; this step was repeated twice. A transmission electron microscope LIBRA 200 FE HR (Zeiss) was used to visualize EVs/EMNVs.

### 2.5. Dynamic Light Scattering

A Malvern Zetasizer NanoZS instrument (Malvern, Worcestershire, UK) was used for the dynamic light scattering (DLS) analysis of the produced EVs and EMNVs. Each EV/EMNV preparation, diluted 1/1000 in PBS filtered through a 20 µM filter (Corning), was analyzed 5 times; 1.5 mL of diluted preparations was loaded into polystyrene cuvettes (DTS0012; Malvern). Analyses were performed at 25 °C (100 measurements) using a 20 mW helium/neon laser (633 nm). Data were analyzed in Zetasizer software 8.01.4906 (Malvern). Ζ-potentials were analyzed in U-type cuvettes (DTS1070; Malvern) with gold electrodes. Measurements of Ζ-potentials were performed at 25 °C at least 5 times. The background signal was measured in filtered PBS.

### 2.6. DOX Packaging

EVs and EMNVs (1 mg of total protein) were loaded by incubation with 400 µg/mL of DOX at 37 °C for 2 h shaking at 500× rpm. Free DOX was discarded by ultrafiltration using via Centrisart 1 (300 kDa) Concentrator (Sartorius) followed by 3 washes with PBS. To estimate the DOX packaging efficiency, optical densities in samples were measured at 480 nm using a spectrophotometer, and the amount of loaded DOX was determined using a calibration curve. Encapsulation efficiency (%) was calculated by the following formula: (DOX incapsulated in vesicles)/(initial amount of DOX used for packaging) × 100.

### 2.7. Western Blotting

EV/EMNV preparations or HEK-293T cells were lysed using 50 μL of RIPA buffer (50 mM Tris-HCl [pH 8.0], 150 mM NaCl, 0,1% Triton X-100, 0.5% sodium deoxycholate, 0.1% sodium dodecyl sulphate [SDS], 1 mM NaF) and incubated with agitation for 30 min at 4 °C followed by sonication for 30 seconds. Samples were loaded with 6× Laemmli buffer (5:1, 60 µg/well) onto 10% SDS-PAAG and then transferred on nitrocellulose membrane. Membrane was blocked with 5% milk in TBS-T (20 mM Tris, pH 7.5, 150 mM NaCl, 0.1% Tween 20) and stained with primary antibodies to exosome markers (EXOAB-KIT-1 for CD63, CD9, CD81 and Hsp70, SBI) 1:1000 or to β-actin (A1978, Sigma) 1:5000 in 5% milk in TBS-T overnight at 4 °C. Membranes were washed 3× with TBS-T and incubated for 1 h with goat anti-rabbit HRP-conjugated antibodies (Ab6721, Abcam, Cambridge, MA, USA) or with goat anti-mouse HRP-conjugated antibodies (Ab6787; Abcam) diluted 1:5000 in 5% milk in TBS-T. Membranes were washed 3× with TBS-T and chemiluminescent signal was developed with SuperSignal West Femto Maximum (Thermo-Fisher) and detected with an X-ray film with 2 h exposure.

### 2.8. Flow Cytometry Analysis

At harvest, cells were analyzed on a BD FACSCanto II flow cytometer (BD Biosciences, San Jose, CA, USA). Briefly, cell culture medium was discarded, and cells were washed twice in PBS, detached from the plates in trypsin-EDTA, resuspended in complete medium, and washed twice in PBS. EGFP-positive cells were detected in the FITC channel; DOX-positive cells were analyzed in the PerCIP-Cy5-5 channel. Data were acquired with BD FACSDiva software and analyzed with NovoExpress software version 1.6.1 (ACEA Biosciences, San Diego, CA, USA).

### 2.9. Cytotoxicity Analysis

HEK-293T, HepG2, and SKBR3 cells were seeded onto 96-well plates at ~30% confluency. HEK-293T cells were treated with DOX (500 nM), HCQ (50 µM), or EV- or EMNV-loaded DOX (500 nM) alone or in combination with 50 µM of HCQ for 12 h. HepG2/SKBR3 cells were treated with EV-loaded DOX at a concentration of 100 nM. Afterwards, Cell Cytotoxicity Assay Kit Reagent (ab112118; Abcam) was added to cells according to the manufacturer’s instructions. Optical density was measured using a CLARIOstar Plus Microplate Reader to calculate the relative viability of cells.

### 2.10. Activated Caspase 3 Assay

HEK-293T cells were transfected with plasmid encoding ZipGFP-Casp3 (a gift from Xiaokun Shu, Addgene 81241) [24], which encodes ZipGFP-based TEV protease reporter for apoptosis visualization. Reporter fluorescence was measured with FACS analysis.

### 2.11. LysoTracker Assay

HEK-293T cells were incubated with 50 µM of HCQ in complete FluoroBrite DMEM media for 24 h. Next, LysoTracker Deep Red was added for 1 h and visualized with fluorescence microscopy.

### 2.12. ExoGlow Protein (Green) Staining of NPs

NPs were stained with an ExoGlow™-Protein EV Labeling Kit (Green) (EXOGP300A-1, SBI) as described by the manufacturer with modifications. Briefly, 500 µg of NPs was stained with the ExoGlow labeling dye and incubated at 37 °C in a shaking incubator at 350× rpm for 20 min; then, free ExoGlow dye was discarded via ultrafiltration using a Centrisart 1 (300 kDa) concentrator (Sartorius) followed by 3 washes with PBS.

### 2.13. High-Content Fluorescent Live-Cell Imaging

HEK-293T cells were seeded into Cellvis 96-well black glass-bottom plates (Thermo Fisher Scientific) at ~30% cell density, treated with DOX-loaded or ExoGlow-labeled NPs, and analyzed with a CELENA^®^ X High-Content Imaging System at 2–22 h post-addition of NPs. The internalization of NPs was measured as the mean fluorescence of ExoGlow-labeled NPs; DOX release was quantified using the mean fluorescence of intracellular DOX. The analysis was performed in Leica LAS X Life Science software version 3.7.6 (Leica Microsystems, Wetzlar, Germany).

### 2.14. Statistical Analysis

Values were expressed as the mean ± standard deviation (SD) in GraphPad Prism 7.0 software. Student’s *t*-test or one-way ANOVA, where applicable, with Tukey’s HSD post hoc test was used to compare variables and calculate *p*-values to determine statistically significant differences in means.

## 3. Results and Discussion

### 3.1. Physical and Biological Properties of the Biological Vesicles

First, we characterized the physical and biological properties of HEK-293T cell-derived EVs and EMNVs (Figure 1) as described in the Materials and Methods. TEM analysis showed that EVs and EMNVs were nanoparticles clearly defined by a proteolipid membrane (Figure 1a,b). EMNVs had a regular shape, while EVs showed some aggregation and deviation from a spherical shape (Figure 1a insets). This phenomenon has already been highlighted for EVs [27], while extrusion is likely the reason for the more regular shape of EMNVs. Size distribution analysis (Appendix A) corroborated TEM analysis, showing a similar polydispersion index for both EVs and EMNVs. DLS (Figure 1c) demonstrated that the vesicles ranged in size between 120 and 140 nm (Figure 1c). Additionally, the vesicles were negatively charged (Figure 1d) with EMNVs showing a stronger negative surface charge than EVs (−22 mV vs. −15 mV). Due to the presence of proteins, a negative charge was expected in both the proteolipid vesicles; the lower negative charge of EMNVs can be explained by a higher concentration of proteins on the surface of these vesicles, different protein density, or charge redistribution [28,29]. The biological properties of the vesicles were defined by investigating four biomarkers currently representing the dogma [30] of biological particle characterization, including CD81, CD9, Hsp70, and CD63 (Figure 1e). All these biomarkers were detectable and similar in band intensity in both the vesicles. β-actin was expressed in HEK-293T cell lysates, and, expectedly, was undetectable in NP lysates. These data demonstrated that our synthetic route to produce EMNVs supported the generation of a pure product, very similar to EVs. 

### 3.2. Delivery Properties of DOX-Loaded EVs and EMNVs, Alone or in Combination with HCQ

In terms of synthesis, the recovery yield (measured by total protein) of EMNVs was 60-fold higher than EVs secreted from the same number of HEK-293T cells (Figure 2a). When the vesicles were passively loaded with DOX, loading yield was similar in both types of particles (Figure 2b). At this stage, we tested the effects of HCQ on the endosomal compartment of HEK293T. As shown in Figure 2c depicting LysoTracker red accumulation before and after HCQ treatment, this molecule decreased the probe’s fluorescence intensity, in line with other papers showing HCQ activity in destabilizing the cellular endosomal compartment and ability to enhance the cytoplasmic drug release [31]. To directly compare EVs’ and EMNVs’ drug delivery properties, HEK293T cells were treated with free chemotherapeutic or DOX-loaded NPs. These treatments were also performed in the presence of HCQ to determine the effect of this drug on DOX cellular accumulation. DOX internalization was evaluated via flow cytometry and both the percentage of fluorescent cells and mean fluorescence intensity measures were analyzed (Figure 2d and Appendix A). Negative controls were represented by untreated cells and cells treated with HCQ alone to verify that these treatments did not increase cell fluorescence in the investigated channel.

Compared to untreated cells and negative controls, DOX-loaded EVs and EMNVs resulted in a similar percentage of positive cells (over 90%) (Figure 2d). However, when the mean fluorescence intensity was analyzed, the same samples showed a different internalization profile. In particular, combinatorial treatments of loaded particles with HCQ significantly increased the fluorescence intensity of DOX-loaded NPs (Appendix A), compared to free-DOX and DOX-loaded particles alone. This could be explained by the auto-quenching effect of DOX, occurring when vesicles are compartmentalized in the endosomal vesicles rather than after diffusion in the cytoplasm [32]. However, from these data, we could not exclude that HCQ directly increased the particle internalization rate. To precisely analyze this phenomenon, we performed timelapse (from 2 to 22 h post-treatment) analysis of DOX release from EVs and EMNVs with and without HCQ using high-content fluorescent imaging (Figure 2e). In this assay, EVs were consistently, but not statistically significantly, more efficient in increasing DOX-derived cellular fluorescence compared to EMNVs (Figure 2e). The benefits of HCQ in increasing DOX fluorescence were evident also at the early time points of treatment, indicating that this molecule could increase the particle internalization rate. The increased release of DOX is likely attributed to the increased endolysosomal escape reported for HCQ at other systems [20,21].

For this reason, we labeled EVs/EMNVs with ExoGlow, a fluorescent dye that was equally effective in labeling both types of NPs, to investigate the internalization kinetics of the carriers using fluorescent microscopy. We observed higher internalization of EMNVs compared to EVs (Figure 2f,g). To the best of our knowledge, this is the first report demonstrating higher internalization of EMNVs compared to EVs. Most importantly, for the first time, here, we report a consistent increase in EV (* *p* < 0.05) and EMNV (ns, *p* > 0.05) internalization in combination with HCQ (Figure 2f,g). Interestingly, when the particles were administered in the absence of HCQ, the internalization rate appeared to decrease slightly. This phenomenon could be explained by cell division (particularly pronounced in HEK-293T) halving the particle content at every cycle [33].

These data were corroborated by the analysis of cell viability performed after 57 h of treatment, demonstrating that, although HCQ alone neither had cytostatic effects nor increased free DOX cytotoxicity, it enhanced the efficacy of DOX encapsulated in both the vesicles (Figure 2h) in a similar fashion. The HCQ ability in increasing the toxicity of loaded vesicles was confirmed also in human hepatoma (HepG2) and breast cancer (SKBR3) cells, as shown in Appendix A. Finally, to evaluate if the two systems had similar toxicity mechanisms, we tested the activation of caspase 3 [24] as a DOX-induced apoptosis marker. This analysis is also important to exclude the potential perturbation of cell viability by HCQ at the molecular level, as the activation of caspase 3 has been observed in cells treated with toxic doses of HCQ [34]. DOX or DOX-loaded NPs consistently activated this apoptotic mediator in all the samples delivering the chemotherapeutic, suggesting that EVs and EMNVs work similarly, while HCQ did not induce the activation of caspase 3 (Figure 2i).

## 4. Conclusions

In this work, we directly compared the performance of EVs and EMNVs in delivering DOX. Despite the similar biological and physical properties, EMNVs’ internalization was slightly more effective than EVs’ and the particles showed similar cytotoxic effects (Figure 2e–g). Importantly, for the first time, we demonstrated the effect of the FDA-approved drug HCQ in increasing the internalization of EVs/EMNVs and their efficacy in killing cancer cells (Figure 2h). This is the first indication that a lysosome-destabilizing agent can improve the performance of biological NPs.

More investigation is necessary to determine if these benefits persist in different experimental scenarios (i.e., the ability to encapsulate other therapeutics, in vivo biodistribution, targeting, efficacy, and side effects) and identify the mechanism at the basis of increased particle internalization in combination with HCQ. Finally, it is important to highlight that EMNVs can serve as a delivery model in many studies, providing a cost-effective and a highly reproducible tool to investigate biomimetic particle properties.

## Figures and Tables

**Figure 1 pharmaceutics-15-00534-f001:**
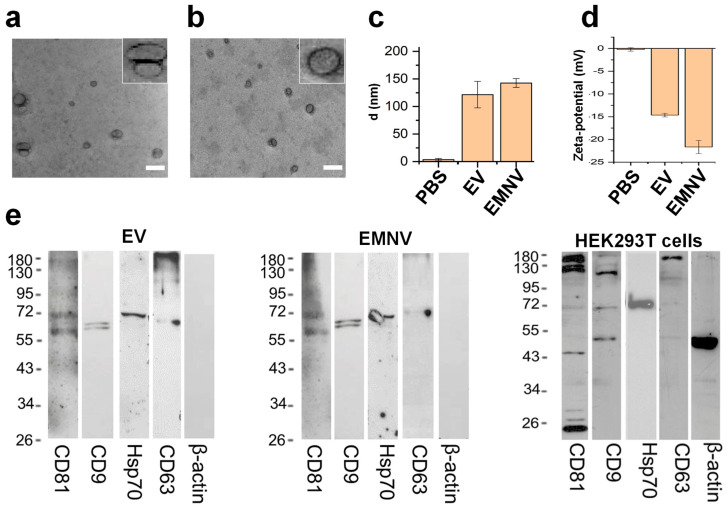
Characterization of EVs and EMNVs. Transmission electron microscopy images of (**a**) EVs and (**b**) EMNVs with an enlarged image in the inset (scale bar = 200 nm). Dynamic light scattering analysis of EV/EMNV (**c**) mean diameter, and (**d**) Ζ-potential. (**e**) Western blot analysis of expression of CD81, CD9, Hsp70, CD63, and β-actin proteins in preparations of EVs and EMNVs and in HEK-293T cell lysates.

**Figure 2 pharmaceutics-15-00534-f002:**
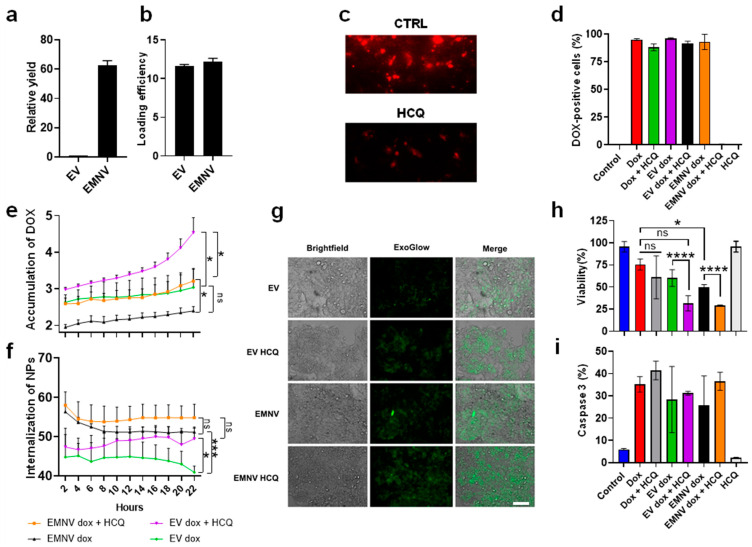
Internalization and therapeutic properties of EVs and EMNVs alone or in combination with HCQ. (**a**) Relative yield of EVs and EMNVs. (**b**) Efficiency of DOX loading into EVs and EMNVs. (**c**) LysoTracker analysis of lysosome destabilization by HCQ (reduction in the LysoTracker signal in HCQ group). (**d**) Percentage of DOX-positive cells. (**e**) Release of encapsulated DOX (intracellular accumulation) in EVs and EMNVs with and without HCQ. (**f**) Internalization of ExoGlow-labeled EVs and EMNVs with and without HCQ. (**g**) Representative images of HEK-293T cells treated with ExoGlow-labeled NPs. Scale bar = 100 µm. (**h**) Analysis of HEK-293T cell viability with different treatments 24 h post-treatment. (**i**) Percentage of activated-caspase-3-positive cells. * *p* < 0.05; *** *p* < 0.001; **** *p* < 0.0001, ns—not significant.

## Data Availability

All data available are presented in the body of the manuscript.

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
