# Peer review of "Hydroxychloroquine Enhances Cytotoxic Properties of Extracellular Vesicles and Extracellular Vesicle–Mimetic Nanovesicles Loaded with Chemotherapeutics"

_pharmaceutics, 2023, doi:10.3390/pharmaceutics15020534_

Round 1

Reviewer 1 Report

An article entitled ‘’Hydroxychloroquine enhances cytotoxic properties of extracel- 2 lular vesicles and extracellular vesicle-mimetic nanovesicles  loaded with chemotherapeutics’’ by  Brezgin et al. reveal that at least in vitro, the use of synthetic biomimetic particles is comparable to the 43 natural counterparts, while their synthesis is significantly faster and more cost effective. In addition, 44 we highlighted the benefits of combining biological nanoparticles with a lysosome destabilizing 45 agent that increased the delivery properties of the particles. The finding is interesting, however the abstract section should be improved. I provided some comments on the manuscript. Concerns should be addressed by authors.

1.     Some English typo errors must to be corrected.

2.     Rephrase and better the abstract regarding results.

3.     Authors cite these references in introduction (PMID: 35427569; PMID: 33246476; https://doi.org/10.1016/j.matdes.2020.109227).

4.     What is the main idea for preparing EMNVs instead of producing EVs/exosomes from cells? As you know, EVs have many potentials such as homing and immune escape properties which make them ideal for smart drug delivery.

5.     HCQ inhibits autophagy. Exosomes biogenesis links autophagy signaling at molecular and vesicular levels. Two pathways may adjust each other, therefore this is important to evaluate them. What is correlation in your study?

6.     Authors use cancer cells to validate results.

7.     Why HCQ did not reduce cell viability and increase apoptosis? (Fig. 2 a and b).

8.     Authors investigate uptake test for EVs and EMNVs.

Author Response

We would like to cordially thank the Reviewers for the highly valuable and professional evaluation of our study. The issues raised by the Reviewers, and now answered in our revised manuscript, substantially improve its quality and the validity of the data presented. To answer the critiques, we have performed three additional experiments on internalization of EVs/EMNVs and intracellular release of DOX from EVs/EMNVs using high content fluorescent live cell image analysis as well we reproduced the most important data in two cancer cell lines (SKBR3 and HepG2). The precise responses to the critiques follow:

Reviewer #1, 1: Some English typo errors must to be corrected.

Authors’ response #1: Thank you, the manuscript was revised by a professional, native-speaking English editor.

Reviewer #1, 2: Rephrase and better the abstract regarding results.

Authors’ response #1, 2: We thank the Reviewer for this suggestion. The abstract was revised for clarity, the most important results of the study were added.

Reviewer #1, 3: Authors cite these references in introduction (PMID: 35427569; PMID: 33246476; https://doi.org/10.1016/j.matdes.2020.109227).

Authors’ response #1, 3: We are grateful to the Reviewer for this suggestion. Unfortunately, the «Communications» type of the manuscript requires to be very concise, and we have to miss many brilliant references to include only the most relevant and important. As such, we would prefer to leave the reference list as is.

Reviewer #1, 4: What is the main idea for preparing EMNVs instead of producing EVs/exosomes from cells? As you know, EVs have many potentials such as homing and immune escape properties which make them ideal for smart drug delivery.

Authors’ response #1, 4: The major advantage of EMNVs is that «unlike EV production, EMNV manufacturing does not require long-term cell culturing and large amounts of cell culture media. Their synthesis is independent from the cell state and their synthetic yield (particle/cell ratio) remarkably exceeds that of EVs [14], [15].». We have re-organized the introduction to better highlight the differences, advantages and disadvantages of both types of nanovesicles.

Reviewer #1, 5: HCQ inhibits autophagy. Exosomes biogenesis links autophagy signaling at molecular and vesicular levels. Two pathways may adjust each other; therefore, this is important to evaluate them. What is correlation in your study?

Authors’ response #1, 5: In our study, we used HCQ for treating target cells with two types of nanoparticles to, potentially, increase their cytotoxic properties. HCQ was not used during production of either EVs or EMNVs. These procedures are highlighted in the body of the manuscript and in the materials and methods section.

Reviewer #1, 6: Authors use cancer cells to validate results.

Authors’ response #1, 6: Thank you for this valuable suggestion. We have performed additional experiments in SKBR3 breast cancer cell line and HepG2 human hepatoma cell line, and were able to reproduce our key results at both cancer cell models (please, see in Figure S3). In these experiments, HCQ remarkably enhances cytotoxicity of DOX-loaded EVs.

Reviewer #1, 7: Why HCQ did not reduce cell viability and increase apoptosis? (Fig. 2 a and b).

Authors’ response #1, 7: HCQ is an FDA-approved drug which is used in the clinic as an antimalarial drug, a therapy for rheumatoid arthritis and more. We used a physiologically relevant concentration of HCQ, which was previously reported to induce endolysosomal escape, and, expectedly, observed no detectable toxicity.

Reviewer #1, 8:  Authors investigate uptake test for EVs and EMNVs.

Authors’ response #1, 8: Thank you for the astute suggestion. We have performed an additional experiment using labeled nanoparticles and analyzed internalization of two types of nanoparticles with and without HCQ dynamically from 2 to 22 hours post treatment using high content live cell image fluorescent microscopy. The results for the first time indicated slightly higher internalization of EMNVs compared to EVs (Figure 2f), and demonstrated that HCQ can significantly increase internalization of EVs and consistently, but not significantly, increase internalization of EMNVs. This is indeed a very useful suggestion that substantially improved the quality of our manuscript and demonstrated a new mechanism how HCQ could improve cytotoxic activity of biological nanoparticles.

Reviewer 2 Report

This manuscript compares the drug delivery capacity between natural extracellular vesicles (EVs) and synthetic extracellular vesicle-mimetic nanovesicles (EMNVs), and studies the effect in combination of hydroxychloroquine (HCQ), which is an interesting work. The writing and data presentation are generally good, but still have some errors. Comparison of EVs and EMNVs has been reported in many reviews, but directly comparison in an experimental study is seldomly seen. However, drug delivery studies of EVs and EMNVs have been widely reported (i.e., for doxorubicin (DOX) delivery). The novelty thus seems not high. Some issues need to be addressed, as shown below.

1. Application of EVs and EMNVs for drug delivery (i.e., for doxorubicin (DOX) delivery) have been widely reported. Directly comparison in a same study seems interest but not very novel. Please highlight the novelty of the study in the main text.

2. The manuscript claims “hydroxychloroquine enhances cytotoxic properties”. However, systemic studies on the effect are missing, which hardly support the conclusion.

3. What is the mechanism of “hydroxychloroquine enhances cytotoxic properties”? It seems the data presented are not sufficient to prove it. 

4. What is the purpose of investigating the activation of Caspase-3? Why Dox results in a low MFI similar to HCQ, but DOX+HCQ has a higher value? (figure 2h).

5. “The yield of EMNVs was 60-fold higher than EVs secreted from the same amount of HEK293T cells.” Please specify how to calculate the yield. 

6. Line 238 “showing a similar polydispersion index for both types of nanoparticles”. In fact, it seems the PDI of the two types of nanoparticles is different. Please show the value. By the way, why are not the EMNVs uniform in size? This is not likely the case considering the production process of EMNVs.

7. In figure 2, what was the amount of DOX used in all the test? Please clarify.

8. There are some errors in the writing. Please check and revise.

Author Response

We would like to cordially thank the Reviewers for the highly valuable and professional evaluation of our study. The issues raised by the Reviewers, and now answered in our revised manuscript, substantially improve its quality and the validity of the data presented. To answer the critiques, we have performed three additional experiments on internalization of EVs/EMNVs and intracellular release of DOX from EVs/EMNVs using high content fluorescent live cell image analysis as well we reproduced the most important data in two cancer cell lines (SKBR3 and HepG2). The precise responses to the critiques follow:

Reviewer #2, 1: Application of EVs and EMNVs for drug delivery (i.e., for doxorubicin (DOX) delivery) have been widely reported. Directly comparison in a same study seems interest but not very novel. Please highlight the novelty of the study in the main text.

Authors’ response #2, 1: Thank you for this constructive criticism, we have now highlighted the importance of our study, especially given the new results that we obtained in additional experiments.

In particular, in the abstract of the study we added the following:

«In this study, we investigated the physical, biological, and delivery properties of EVs and EMNVs and analyzed their ability to deliver chemotherapeutic drug doxorubicin. EMNVs and EVs exhibit similar properties, but EMNVs are more effectively internalized, while EVs show higher intracellular doxorubicin release activity. In addition, these nanotherapeutics were investigated in combination with the FDA-approved drug hydroxychloroquine (HCQ). We demonstrate that HCQ induced lysosome destabilization and could significantly increase nanoparticle internalization, doxorubicin release and cytotoxicity. Altogether, these data demonstrate that, from the de-livery standpoint in vitro, the internalization of EMNVs and EVs and their payload release are slightly different and both nanotherapeutics had comparable cytotoxic performances

New sections have also been added to the Results and Discussion sections.

The key findings of our study include (1) evidence that EMNVs are more effectively internalized, while EVs show higher intracellular DOX release activity; (2) effects of HCQ on increased internalization of NPs, (3) increased DOX release and (4) increased cytotoxicity.

Reviewer #2, 2: The manuscript claims “hydroxychloroquine enhances cytotoxic properties”. However, systemic studies on the effect are missing, which hardly support the conclusion.

Authors’ response #2, 2: Indeed, testing the effects of HCQ on cytotoxic properties in vivo is important. However, it goes beyond the scope of this study, where we focus primarily on the effects of HCQ on the effects of biological nanoparticles at the cellular and intracellular levels.

Reviewer #2, 3: What is the mechanism of “hydroxychloroquine enhances cytotoxic properties”? It seems the data presented are not sufficient to prove it.

Authors’ response #2, 3: We concur. To address this critique, we have performed two additional experiments. First, we tracked internalization of labeled nanoparticles with and without HCQ every 2 hours from 2 hour to 22 hour post treatment. This experiment revealed the differences in internalization efficiency of EVs and EMNVs. Moreover, it demonstrated that HCQ could substantially increase internalization of biological nanoparticles (Figure 2f,g). Second, using high content fluorescent live cell imaging we performed timelapse analysis of DOX intracellular release showing the benefits of HCQ in increasing DOX fluorescence released from both EVs and EMNVs.

Altogether, we conclude that

«More investigation is necessary to determine if these benefits persist in different experimental scenarios (i.e., ability to encapsulate other therapeutics, in vivo biodis-tribution, targeting, efficacy, and side effects), and the mechanism at the basis of particle internalization increase in combination with HCQ

Reviewer #2, 4: What is the purpose of investigating the activation of Caspase-3? Why Dox results in a low MFI similar to HCQ, but DOX+HCQ has a higher value? (figure 2h).

Authors’ response #2, 4: caspase-3 activation is one of the mechanisms of DOX-induced cell death and, more importantly, caspase-3 is activated by toxic doses of HCQ (10-75 ug/mL) (Laurence Lagneaux et al., 2022 Leuk Lymphoma). Therefore, applying this analysis we have confirmed activation of at least one signaling pathway related to DOX-mediated cell death and performed a control experiment to exclude the possibility of HCQ-mediated perturbation of cell viability at the molecular level. This is an important control experiment.

We have added the following:

«Finally, to evaluate if the 2 systems had similar toxicity mechanisms we tested the ac-tivation of caspase 3 [24] as DOX-induced apoptosis marker. This analysis is also im-portant to exclude potential perturbation of cell viability by HCQ at the molecular level, as ac-tivation of caspase 3 is observed in cells treated with toxic doses of HCQ [34]. DOX or DOX-loaded NPs consistently activated this apoptotic mediator in all the sam-ples delivering the chemotherapeutic, suggesting that EVs and EMNVs work similarly, while HCQ did not induce activation of caspase 3 (Figure 2i).»

Reviewer #2, 5: “The yield of EMNVs was 60-fold higher than EVs secreted from the same amount of HEK293T cells.” Please specify how to calculate the yield.

Authors’ response #2, 5: Thank you. We have added: «(measured by total protein)». The procedure is described in the materials and methods section.

Reviewer #2, 6: Line 238 “showing a similar polydispersion index for both types of nanoparticles”. In fact, it seems the PDI of the two t1qypes of nanoparticles is different. Please show the value. By the way, why are not the EMNVs uniform in size? This is not likely the case considering the production process of EMNVs.

Authors’ response #2, 6: In DLS, the light scattering intensity is proportional to the sixth power of particle hydrodynamic radius, so that larger particles generate more intensive signal (Varga et al., 2014 J Extracel Vesicles). Considering differences in radius of particles between two populations of EMNVs (~1.5-fold), the sample contains nearly 11-fold more particles of ~120-140 nm hydrodynamic radius, and this population is highly predominant. During EMNVs preparation, filters with 10,5 and 1 um pores are routinely used (described in previous publications). In the extrusion process, particles with sizes less than 200 nm are generated and not eliminated during purification (centrifugation at 10,000 g or ultrafiltration). As these data are hard to interpret without specialized knowledge in the field, so we moved these data to supplementary information.

Reviewer #2, 7: In figure 2, what was the amount of DOX used in all the test? Please clarify.

Authors’ response #2, 7: We have added this information to the materials and methods section:

«HEK-293T, HepG2 and SKBR3 cells were seeded onto 96-well plates at ~30% con-fluency. HEK-293T cells were treated with DOX (500 nM), HCQ (50 µM), or EV- or EMNV-loaded DOX (500 nM) alone or in combination with 50 µM of HCQ for 12 h. HepG2/SKBR3 cells were treated with EV-loaded DOX at a concentration of 100 nM. Afterwards, Cell Cytotoxicity Assay Kit Reagent (ab112118; Abcam) was added to cells according to manufacturer’s instructions. Optical density was measured using CLAR-IOstar Plus Microplate Reader to calculate relative viability of cells.»

Reviewer #2, 8: There are some errors in the writing. Please check and revise.

Authors’ response #2, 8: The manuscript was revised by a professional, native-speaking English editor.

Round 2

Reviewer 1 Report

Now, it is acceptable for publishing.